# Utility of Fetal Echocardiography with Acute Maternal Hyperoxygenation Testing in Assessment of Complex Congenital Heart Defects

**DOI:** 10.3390/children10020281

**Published:** 2023-01-31

**Authors:** Sheetal R. Patel, Nitin Madan, Pei-Ni Jone, Mary T. Donofrio

**Affiliations:** 1Division of Pediatric Cardiology, Ann & Robert H Lurie Children’s Hospital of Chicago, Northwestern University Feinberg School of Medicine, Chicago, IL 60611, USA; 2Children’s Mercy Kansas City, University of Missouri-Kansas City School of Medicine, Kansas City, MO 64108, USA; 3Children’s National Hospital, George Washington University, Washington, DC 20010, USA

**Keywords:** fetal echocardiography, maternal hyperoxygenation test, congenital heart defects, delivery planning

## Abstract

Fetal echocardiography is an excellent tool for accurately assessing the anatomy and physiology of most congenital heart defects (CHDs). Knowledge gathered from a thorough initial fetal echocardiogram and serial assessment assists with appropriate perinatal care planning, resulting in improved postnatal outcomes. However, fetal echocardiography alone provides limited information about the status of the pulmonary vasculature, which can be abnormal in certain complex CHDs with obstructed pulmonary venous flow (hypoplastic left heart syndrome with restrictive atrial septum) or excessive pulmonary artery flow (d-transposition of the great arteries, usually with a restrictive ductus arteriosus). Fetuses with these CHDs are at high risk of developing severe hemodynamic instability with the immediate transition from prenatal to postnatal circulatory physiology at the time of birth. Adjunctive use of acute maternal hyperoxygenation (MH) testing in such cases can help determine pulmonary vascular reactivity in prenatal life and better predict the likelihood of postnatal compromise and the need for emergent intervention. This comprehensive review discusses the findings of studies describing acute MH testing in a diverse spectrum of CHDs and congenital diagnoses with pulmonary hypoplasia. We review historical perspectives, safety profile, commonly used clinical protocols, limitations, and future directions of acute MH testing. We also provide practical tips on setting up MH testing in a fetal echocardiography laboratory.

## 1. Background

Congenital heart defects (CHDs) account for the most common congenital abnormality, occurring in one out of one hundred and ten live births [1,2]. Fortunately, prenatal detection of CHDs by fetal echocardiography is becoming increasingly common [3,4,5] due to advances in diagnostic imaging and improved obstetrical screening guidelines. Fetal echocardiography can evaluate fetal cardiac anatomy and the progression of the disease. Serial assessment throughout the pregnancy with late gestation fetal echocardiography can help determine the anticipated hemodynamic changes after birth and guide perinatal management. This accurate prenatal assessment with individualized perinatal management plans has improved in utero and postnatal outcomes for CHDs [6,7,8]. Although most newborns with CHDs are stable soon after birth, some babies with specific CHDs have severe hemodynamic instability immediately after the placental separation as the fetal circulation starts transitioning to postnatal circulation. Survival in these newborns with critical CHDs requires immediate stabilization in the delivery room along with lifesaving catheter or surgical interventions within the first few hours of life. Examples of these conditions include hypoplastic left heart syndrome (HLHS) with a restrictive or intact atrial septum or d-transposition of the great arteries (d-TGA), most often with a restrictive ductus arteriosus. In babies with these CHDs, opening the atrial septum with balloon atrial septostomy (BAS) is often needed. Another example of a critical CHD requiring urgent cardiac surgical intervention is obstructed total anomalous pulmonary venous return (TAPVR). Accurate prediction of neonatal hemodynamic instability can be challenging using standard fetal echocardiography due to an inability to predict changes from prenatal to postnatal circulation. The addition of maternal hyperoxygenation (MH) testing to standard fetal echocardiography has been shown to improve the accuracy of predictions related to postnatal hemodynamics, thus providing an opportunity for necessary resource planning for anticipated lifesaving cardiac interventions [9,10,11,12,13]. The aim of this review, therefore, is to discuss prior studies evaluating the utility and advantages of fetal echocardiography along with acute MH testing in the management of newborns with complex CHDs at risk of needing urgent cardiac interventions soon after birth.

### 1.1. Fetal Circulation and Transition at the Time of Birth

The normal fetal blood flow pattern (Figure 1A) is characterized by “parallel circulation”, which significantly differs from “in-series” circulation seen after birth (Figure 1B). This parallel fetal circulation is due to three fetal shunts, including ductus venosus, foramen ovale, and ductus arteriosus. The ductus venosus brings the nutrient and oxygen-rich blood from the umbilical vein to the right atrium via the inferior vena cava. The foramen ovale allows for the distribution of nearly half of this systemic venous return to the left side of the fetal heart. The left ventricle (LV) output is mostly distributed to the coronary arteries and the upper part of the fetal body via three branches that originate from the aortic arch. In contrast, most of the output from the right ventricle (RV) passes through the ductus arteriosus to be distributed to the lower part of the fetal body and then back to the placenta via the umbilical arteries. This preferential flow from the RV to the ductus arteriosus is due to the high pulmonary vascular resistance resulting in only a small percentage of the RV output being sent to the branch pulmonary arteries (PAs). Therefore, in fetal circulation, the right and left ventricles are “parallel” to each other, with both ventricles handling a part of the combined cardiac output, and a very small amount going to the lungs.

At the time of birth, dramatic changes occur, in the transition from prenatal circulation to postnatal circulation. Systemic vascular resistance increases due to the placental separation, and pulmonary vascular resistance decreases due to spontaneous respiration and pulmonary vasodilation promoted by increased oxygenation. In addition, increased oxygenation leads to constriction of the ductus arteriosus. These changes cumulatively increase pulmonary blood flow, pulmonary venous return, and left atrial pressures, closing the foramen ovale. With the closure of the fetal shunts, including the foramen ovale, ductus arteriosus, and ductus venosus, the circulatory transition is completed such that the postnatal flow through the right and left heart is “in-series” [15]. Due to the differences between fetal and postnatal circulation, accurately predicting the postnatal hemodynamic effects of CHDs is challenging using standard fetal echocardiography alone. Addition of MH testing by giving 100% oxygen to the mother via a non-rebreather mask for 10 to 15 min partially mimics these postnatal circulatory changes in the fetal circulation, allowing for more accurate prediction of hemodynamic instability following birth as described below. 

### 1.2. Historical Perspective on Maternal Hyperoxygenation Testing

Acute MH testing has been studied for over five decades. Studies have focused on evaluating hemodynamic changes to increased circulating oxygen content in the fetal blood, mimicking the postnatal circulatory physiology. Bertolizio et al. [16] and Frangipani et al. [17] published early reports of the effect of MH on amniotic fluid acid–base equilibrium in 1966 and 1969, respectively. However, the cardiac and circulatory changes with MH were not easy to evaluate prior to the availability of fetal echocardiography in the late 1980s. One of the early studies evaluating the cardiac effects of short-term MH reported that abnormal E/A ratio across the mitral and tricuspid valve inflow Doppler patterns seen in growth-restricted fetuses could be improved with MH [18]. Another early study by Soregaroli et al. [19] published in 1993 reported increased peak flow velocities in ductus venosus after MH but no effect on fetal heart rate. This effect of MH on fetal circulation was more obvious in the third trimester compared to early gestational age [20]. A randomized study published in 1997 by Ramner et al. [21] reported that reduction in the pulmonary vascular impedance (measured as pulsatility index in proximal and distal right and left PAs) with MH is significant at 31–36 weeks of gestation but not at 20–26 weeks of gestation. A more detailed evaluation of the fetal echocardiographic findings in the same cohort by Rasanen et al. [22] further characterized this pulmonary vasoreactivity. With acute MH in late gestation, increased pulmonary blood flow was suggested by a reduction in the pulsatility index (PI) in the branch Pas, and a reduction in ductus arteriosus flow was suggested by an increase in the PI of ductus arteriosus. In addition, there was a reduction in flow across the foramen ovale. All of these circulatory changes returned to baseline after MH was discontinued. Again, these changes were observed only in late gestation (31–36 weeks of gestation) fetuses and not in early pregnancy. This development of pulmonary vasoreactivity has been attributed to the smooth muscle development in the fetal PAs during late gestation [23]. Together, these early studies suggested that acute MH could temporarily mimic the postnatal changes in fetal circulation and paved the way to study the utility of MH in guiding postnatal resource preparation in complex CHDs. Additionally, these studies showed that all changes returned to baseline after MH was discontinued, and no untoward side effects were noted in the fetus or the mother, indicating the safety of such testing prenatally. 

## 2. Clinical Maternal Hyperoxygenation Protocol

The physiologic change of increased fetal oxygen concentration via MH is achieved by administering oxygen to a mother in late gestation for a short duration. There is no universally accepted MH protocol, but most studies have given 100% humidified oxygen to expectant mothers via a non-rebreather mask for 8 to 15 min (Table 1). 

Fetal echocardiography, including Doppler blood flow analysis, is performed at baseline, and fetal echocardiographic variables of interest are re-evaluated after the acute MH for 10 to 15 min to assess the changes in blood flow and vascular impedance across various fetal cardiac structures. The fetal echocardiographic variables of interest are based on the individual CHDs. These lesion-specific changes anticipated with MH are described in detail below and summarized in Table 1 and Table 2. 

Based on the MH protocols used in prior studies, 100% humidified oxygen is typically delivered to the mother using a non-rebreather mask, which provides around 60% inhaled oxygen concentration to the mother for 10 min before reevaluating select fetal echocardiographic measures. Although initial studies reported repeat assessment after 10–15 min of recovery, many later studies did not evaluate recovery phase hemodynamics in light of prior reports of complete resolution of the circulatory changes without untoward effects on the mother or the fetus. There is a learning curve with establishing MH testing in a fetal echocardiography laboratory. Consistently obtaining branch PA pulse wave Doppler (PWD) and deriving PI is very important in gathering accurate information from this test. However, repeatability for branch PA PWD can be challenging. Baseline PWD repeated during the same fetal echocardiogram can provide PI values that have more than 10% variability. Here, we share a few tips and tricks to improve the repeatability of branch PA PWD, consistency in measurements to derive accurate PI values, and interpretation of MH testing: (1) Determining the site of obtaining branch PA PWD. Three specific sites for obtaining branch PA PWD have been described by Szwast et al. [9]. It is helpful to practice one site of interrogation at first, and we have noted success at the mid-branch PA level. PI values vary depending on the site of PWD interrogation in the branch PA. Hence, obtaining the PWD at the same site is crucial after MH testing. (2) Keeping the PWD angle of interrogation to <10°. (3) Keeping the same PWD gain and PWD scale and using the same probe pre and post-MH. Since MH testing is typically performed in the late trimester, we use our lower-frequency transducer. (4) When fetal position changes post-MH testing such that PWD cannot be repeated at the same angle, recognizing the expected pattern change in the PWD, not just the PI value itself. Characteristic branch PA PWD signal is spiky with a quick sharp upstroke, short systolic time interval, and absent, or only a small degree of flow in diastole. With MH, the PWD pattern should become wider with more flow in both systole and diastole (Figure 2). (5) Verifying the automatic tracings and, when needed, performing manual measurements to derive PI. Some imaging platforms, such as Philips, can derive PI values from PWD using the high Q automatic Doppler analysis. This method can be erroneous, especially given the low end-diastolic velocities in branch PA PWD. (6) Finally, establishing an internal quality improvement project to reduce variability between repeated PWD PI values to <10% is recommended.

## 3. Expected Findings with Maternal Hyperoxygenation Specific to CHD Lesions

### 3.1. Hypoplastic Left Heart Syndrome (HLHS) 

HLHS is a spectrum of CHD that results in the LV being incapable of providing adequate systemic perfusion. As egress through the left-sided cardiac structures is obstructed in HLHS, having an unobstructed atrial septal defect is necessary for the pulmonary venous return to drain from the left to the right atrium. The restrictive or intact atrial septum (RAS) (Figure 3) is reported to occur in 6–20% of newborns with HLHS [35,36]. These newborns require urgent cardiac intervention to open the atrial septum and allow egress of pulmonary venous blood from the left atrium in order to survive. 

Serial assessment with late gestation standard fetal echocardiography is necessary as restrictive atrial septum may develop later in gestation. It can be detected by late gestation fetal echocardiography demonstrating a small or absent atrial septal opening with concomitant flow reversal in the pulmonary veins secondary to left atrial hypertension [37]. The finding of pulmonary venous Doppler flow pattern with an abnormal ratio of forward to reverse velocity–time integral (F/R VTI) (Figure 3D) is associated with increased risk of compromise from atrial septal restriction after birth [28,29,30,37,38]. Pulmonary vein Dopplers have high positive predictive value for detecting HLHS cases with a severely restrictive or intact atrial septum. However, sensitivity is low given the limited volume of pulmonary blood flow in utero, masking these Doppler abnormalities. MH testing as an adjunct to fetal echocardiography may help improve the sensitivity of fetal echocardiography in predicting RAS. 

Szwast et al. [9] evaluated pulmonary vascular reactivity in 43 fetuses with HLHS in response to acute MH. This study evaluated the Doppler flow pattern and PI (peak systolic velocity—end-diastolic velocity/mean velocity) in the proximal, midportion, and distal branch PAs as measures of vascular impedance. Either the right or left pulmonary artery was selected based on the position of the fetus with an angle of interrogation <10°. Fetal echocardiographic variables were obtained at baseline on room air, after 10 min of MH, and then 5 min after stopping the oxygen during the recovery phase. Fetuses that demonstrated a significant reduction in pulmonary arterial PI, indicating an increase in the pulmonary blood flow in response to MH, had wide open atrial communication after birth. In comparison, fetuses with <10% reduction in PI indicating reduced pulmonary vasoreactivity to MH required immediate intervention on the atrial septum after birth. The authors reported that MH could predict the need for immediate cardiac intervention at birth with 100% sensitivity, 94% specificity, 71% positive predictive value, and 100% negative predictive value. Schidlow et al. [12] conducted a prospective study with MH testing and implications for critical care delivery planning among fetuses with CHDs. Pulmonary vasoreactivity was defined as a reduction in PI by 20% for this study. This study included four fetuses with HLHS with concern for RAS in two and intact atrial septum with a decompressing vertical vein in the remaining two. Three fetuses showed good pulmonary vascular reactivity and did not need urgent postnatal cardiac interventions for atrial septum. One fetus that did not exhibit pulmonary vascular reactivity with MH required BAS. Another study by Enzensberger et al. [11] evaluated MH response in 22 fetuses with HLHS and reported that the degree of changes in lung perfusion (qualitative assessment of color Doppler of PAs and quantitative assessment of PI of pulmonary veins) with short-term MH might be a useful adjunct in assessing pulmonary vasculopathy in this patient population. In addition to the pulmonary vasoreactivity with MH, changes in the pulmonary venous flow Doppler F/R VTI ratio are also helpful in assessing RAS. In a more recent study by Mardy et al. [13], a pulmonary venous F/R VTI ratio of ≤6.5 with MH was a more reliable measure for predicting a need for atrial septal intervention after birth with 100% sensitivity and 100% specificity as compared to the assessment of pulmonary vasoreactivity by using PI in the branch PAs. Cox et al. [27] studied nine fetuses with left heart hypoplasia and nine controls. With MH, the PA PI decreased, suggesting a decrease in pulmonary vascular resistance, and pulmonary vein VTI increased, suggesting an increase in pulmonary venous return similar to the prior studies. Additionally, the LV strain and strain rate worsened, suggesting that these changes were due to an increase in LV afterload (secondary to an increase in cerebrovascular resistance). In contrast, the RV strain and strain rate improved, which was thought to be due to decreased RV afterload (secondary to a reduction in pulmonary vascular resistance). Finally, an abstract published in 2018 by Rychik et al. [26] reported changes in Doppler flow patterns after MH testing in 182 fetuses, 114 of whom had HLHS. There was increased cerebral resistance and reduced pulmonary resistance secondary to pulmonary vasodilation. An increase in ductal flow reversal was noted as more flow was diverted away from the ductus arteriosus and toward the pulmonary circulation, resulting in an increase in the pulsatility index in the ductus arteriosus. More importantly, there was no evidence of ductal constriction or change in ventricular performance, supporting MH as a safe means for provocative testing during fetal life.

In summary, acute MH testing as an adjunct to standard fetal echocardiography for fetal HLHS can improve the accuracy of predicting RAS. Findings from MH testing may facilitate postnatal planning, including resource preparation for emergent cardiac interventions for opening the atrial septal communication. Reduced pulmonary vasoreactivity and pulmonary vein F/R VTI ratio < 6.5 with MH are useful parameters in predicting RAS that would require resource planning for catheter-based or surgical interventions to open up the left atrial egress after birth. If an atrial septal restriction is suspected based on late gestation standard fetal echocardiography with or without MH, delivery planning should include all resources needed for opening up the atrial septum soon after birth.

### 3.2. Total Anomalous Pulmonary Venous Return (TAPVR)

Obstruction of the vertical draining vein connecting the pulmonary venous confluence to the systemic vein may result in significant respiratory distress and hemodynamic instability soon after birth. Data are limited on fetal echocardiographic predictors of obstructed TAPVR. One study reported that a vertical vein Doppler peak velocity of >0.74 m/s was predictive for preoperative pulmonary venous obstruction [31]. However, predicting TAPVR obstruction is challenging prenatally, as fetal circulation allows a minimal amount of pulmonary blood flow, and associated small volume pulmonary venous return may mask the actual obstruction that can become significant after birth once the pulmonary blood flow increases. In these cases, MH is expected to increase the pulmonary blood flow and therefore unmask the obstruction of the vertical vein prenatally. Schidlow et al. [12] reported the use of MH in two cases of TAPVR, and the mean gradient in the vertical vein after MH correlated with the postnatal gradients. In one of the fetuses, the mean gradient across the vertical vein was 2 mmHg at baseline, which increased to 12 mmHg with MH. This fetus developed significant obstruction of the vertical vein with low cardiac output and needed extracorporeal membrane oxygenation (ECMO) support until surgical repair could be performed. The second fetus had a resting gradient of 2 mmHg at the baseline, with an increase in gradient to 5 mmHg with MH. This fetus was stable at birth with a postnatal gradient of 6 mmHg in the vertical vein, which was well tolerated until surgical repair at 3 months of age. Based on this study, MH may prove to be a useful tool in assessing TAPVR obstruction. The fetal echocardiogram that is most useful in making these predictions is one that is performed in late gestation (>35 weeks gestation if possible) and with MH as an adjunct to the standard fetal echocardiography. Delivery planning for suspected TAPVR obstruction should include resource preparation for emergent cardiac surgical intervention to relieve the obstructed TAPVR.

### 3.3. D-Transposition of the Great Arteries with Restrictive Atrial Septum

The newborn with d-TGA is at risk of severe hypoxemia if there is inadequate mixing of the oxygenated and deoxygenated blood at the atrial level due to the parallel nature of the systemic and pulmonary circulation. Therefore, restrictive flow across the foramen ovale will result in severe hypoxemia and acidosis, and emergent BAS is required soon after birth. In addition, newborns with d-TGA are also at risk of severe pulmonary hypertension contributing to extreme hypoxemia. Failure to intervene may result in severe hemodynamic decompensation and poor outcomes, including death [32,39,40]. Prediction of RAS by fetal echocardiography remains challenging due to the low sensitivity and low negative predictive value of fetal echo variables. For example, an early study published in 2004 by Jouannic et al. evaluated 130 fetuses with TGA. Twenty-four of these fetuses had at least one abnormal shunt (23 with RAS, five with abnormal ductus arteriosus flow, and four with both). Thirteen of these newborns had profound cyanosis after birth, and two died despite aggressive resuscitation. Both of these deaths occurred in fetuses with ductal constriction in addition to the RAS. The specificity and sensitivity of fetal echo in predicting neonatal emergencies were 84% and 54%, respectively. However, when both of these structures were abnormal, the specificity increased to 100%. This study suggested that ductal constriction may play a significant role in d-TGA physiology and should be evaluated carefully. In a meta-analysis by Buca et al. from six studies that included 292 fetuses, restrictive appearance of the foramen ovale and hypermobile atrial septum were associated with increased risk of requiring urgent BAS (within 24 h of birth). The mean ratio between the foramen ovale size and that of the aortic valve and the mean ratio between the foramen ovale size and that of the total atrial septal length were significantly smaller in fetuses requiring urgent BAS at birth [32]. Again, fetal echocardiographic findings had high specificity but poor sensitivity in predicting RAS in d-TGA. A more recent study by Masci et al. [41] evaluated 31 fetuses with dTGA and found that fetal echocardiographic variables cannot predict the need for urgent BAS after birth. Slodki et al. [33] reported pulmonary venous maximum velocity of >41 cm/s as a helpful predictor of postnatal need for BAS. Despite multiple features that could suggest a risk of foramen ovale restriction, accurately predicting the need for urgent BAS based on standard fetal echocardiographic findings remains challenging. Therefore, a few studies have evaluated MH as an adjunct to improve this prediction model. Schidlow et al. [12] evaluated four fetuses with d-TGA with fetal echocardiographic concerns for an RAS between 35 and 38 weeks of gestation. In two fetuses, MH testing showed good pulmonary vasoreactivity (defined as a decrease in PI by ≥20%) and antegrade flow in the ductus arteriosus. Testing in one showed predominantly left-to-right flow across the foramen ovale with MH, and that fetus did not need BAS after birth. The second fetus continued to have bidirectional shunting across the foramen ovale even with MH and required BAS after birth. The two additional fetuses that did not show adequate pulmonary vasoreactivity with MH needed BAS after birth. These fetuses also had either restricted or bidirectional shunt across the foramen ovale. Therefore, the study suggested that flow across the foramen ovale with MH may help determine the need for BAS. In addition, the study shows that pulmonary vascular reactivity can be assessed as well, and a lack of reactivity increases the likelihood of postnatal compromise and the need for urgent interventions. Further studies are needed to understand the utility of MH in predicting RAS in d-TGA, the impact of pulmonary hypertension, and the lack of pulmonary vasoreactivity as important factors in the postnatal clinical presentation of these patients. 

Given the limitations of current standard fetal echo findings and limited experience using MH, it is recommended that all newborns with d-TGA, especially with an intact ventricular septum, be considered high risk. The delivery plans for such fetuses should include delivery at or near a cardiac center to facilitate immediate postnatal care. Planned delivery with labor induction after 39 weeks allows for preparation for urgent BAS. A cesarean section may be considered in rare instances with prenatal foramen ovale closure or severely restrictive ductus arteriosus. Neonatal resuscitation and stabilization until the BAS may require resource planning, such as the availability of prostaglandin E1 (PGE1) infusion and mechanical ventilation. Additionally, 100% oxygen and inhaled nitric oxide (iNO) may be beneficial in those babies who are noted to have a restrictive ductus arteriosus in utero and are at risk of pulmonary hypertension resulting in severe postnatal hypoxemia. 

### 3.4. Severe Ebstein Anomaly of the Tricuspid Valve 

Ebstein anomaly is a rare form of CHD that can result in severe tricuspid regurgitation, reduced effective RV size, massive cardiomegaly, heart failure, hydrops, fetal arrhythmias, and lung hypoplasia [42]. Newborns with severe Ebstein anomaly or tricuspid valve dysplasia can experience severe hemodynamic instability soon after birth, with high neonatal mortality as reported in a multicenter study by Freud et al. [43]. The percentage of fetuses with hemodynamic compromise increases as the pregnancy progresses, indicating a need for serial assessment [44]. Severe cases may require ECMO cannulation to allow the pulmonary vascular resistance to fall and hemodynamics to improve. However, predicting the postnatal hemodynamic instability in fetuses with Ebstein anomaly remains challenging. Postnatal survival in these fetuses depends on the ability of the RV to generate antegrade pulmonary blood flow. However, the high pulmonary vascular resistance in utero prevents the assessment of the true functional capacity of the RV and patency of the pulmonary valve, which may be functionally atretic. With MH, transient lowering of pulmonary vascular resistance can help predict the ability of the RV to generate forward flow in the postnatal period. Schidlow et al. [12] evaluated the utility of MH in two fetuses with severe Ebstein anomaly with severe tricuspid regurgitation, trivial antegrade pulmonary blood flow, and retrograde flow in ductus arteriosus. One of these fetuses demonstrated pulmonary vascular reactivity with increased cardiac output across the pulmonary valve. This fetus was successfully managed without PGE1 and iNO with a gradual increase in pulmonary blood flow. The other fetus, who demonstrated reduced pulmonary vasoreactivity and no change in cardiac output across the pulmonary valve, developed a circular shunt, diminished cardiac output, and multiorgan failure, and died at 10 days of age. These observations suggest the potential use of MH testing to stratify risk for fetuses with severe Ebstein anomaly of the tricuspid valve. For fetuses with extreme cardiomegaly, functional pulmonary valve atresia, or circular shunt physiology prenatally, postnatal resource preparation should include the availability of inhaled oxygen and inhaled nitric oxide to promote pulmonary vasodilation as well as ECMO backup in case it is needed. Additionally, if there is a patent pulmonary valve with pulmonary regurgitation, carefully monitoring the newborn without initiation of PGE1 infusion is suggested to reduce the risk of circular shunt physiology. 

### 3.5. Pulmonary Hypoplasia 

MH has been used to predict the severity of pulmonary hypoplasia and the risk of neonatal death in fetuses with congenital anomalies that may cause pulmonary hypoplasia, such as congenital diaphragmatic hernia and congenital pulmonary adenomatous mass. For example, in a 2002 study by Broth et al. [45] evaluated pulmonary vasoreactivity with MH defined as less than a 20% reduction in the PI of the Doppler blood flow within the first branch of either the right or left PA in fetuses with pulmonary hypoplasia. They noted that absence of pulmonary vasoreactivity to MH during prenatal assessment correlated with neonatal death from pulmonary hypoplasia. In comparison, a reactive test (≥20% reduction in PI) predicted >90% of surviving infants. Notably, the use of ECMO was not as widespread for pulmonary hypoplasia when this study was conducted in 2002, which could have affected the mortality. However, MH helped predict postnatal instability and outcomes in fetuses with pulmonary hypoplasia. 

## 4. Maternal and Fetal Safety with Hyperoxygenation

Since the introduction of acute MH testing about five decades ago and fetal echocardiographic evaluation of cardiovascular changes with acute MH followed by 5 to 10 min of recovery, there have been no known significant side effects to the fetus or expectant mothers. For example, a study evaluating the hemodynamic effects of MH showed no statistically significant changes in maternal ventilation, blood pressure, heart rate, oxygenation indices, and Doppler velocity in the maternal internal carotid and uterine artery [46]. Fetal cardiac hemodynamic effects of acute MH are shown to be reversible within 5 to 10 min of stopping MH based on the fetal hemodynamic assessment performed in the recovery phase. Szwast et al.’s study [9] reported no adverse events with MH. In the recovery phase, PA Doppler returned to baseline, and no significant ductal constriction (defined as PI < 1.9) or change in the middle cerebral artery or umbilical artery PI was noted. A systematic review of the utility of MH by Co-Vu et al. reported that acute MH was safe, and three of the included studied noted no untoward effects in the mother or fetus [10]. As published in an abstract [26] reporting the authors’ experience with 182 fetuses (114 with HLHS) who underwent MH, there were no significant changes in the umbilical arterial PI (placental resistance was unchanged), and ventricular mechanics remained unchanged after acute MH, further supporting that acute MH is safe for mother and the fetus. This study reported an increase in the cerebral resistance, but the changes were temporary. These findings suggest that when MH is used as a provocative test, by giving oxygen for a short period as a diagnostic aid, there were no significant adverse events based on the maternal clinical parameters or fetal echocardiography Doppler parameters, indicating that this test is safe for the mothers and fetuses [10]. Therefore, in many centers, MH is no longer considered an experimental test and is used as part of a clinical protocol. Although, data from these prior studies remain limited and further studies in this area would be beneficial in ensuring the safety and utility of MH testing. 

## 5. Limitations and Future Directions

Although MH seems to be a promising adjunct to late gestational standard fetal echocardiography, certain limitations have prevented more widespread use of this technique. First, there is no universal definition of normal vs. abnormal pulmonary vasoreactivity to MH, and different studies have used either a <10% reduction in PA PI or <20% reduction in PA PI to define reduced pulmonary vasoreactivity. Choosing a higher cut-off value can increase the specificity but reduce sensitivity. In addition, the reproducibility of PI assessment can be challenging unless a systematic approach is implemented in making these measurements as described earlier. Although the utility of MH in HLHS is more widely studied, more research studies are needed to understand its utility in other CHD lesions. 

## 6. Conclusions

MH testing as an adjunct to standard fetal echocardiography can provide insight into fetal pulmonary vasculature and help predict the postnatal hemodynamics after birth and the transition of fetal circulation to postnatal circulation. In addition, this test can be helpful in risk stratification for perinatal management of fetuses with complex CHDs with a high risk of hemodynamic instability, such as HLHS with a restrictive or intact atrial septum, d-TGA with an RAS and potentially pulmonary hypertension, TAPVR, Ebstein anomaly of the tricuspid valve, and congenital defects with lung hypoplasia. 

## Figures and Tables

**Figure 1 children-10-00281-f001:**
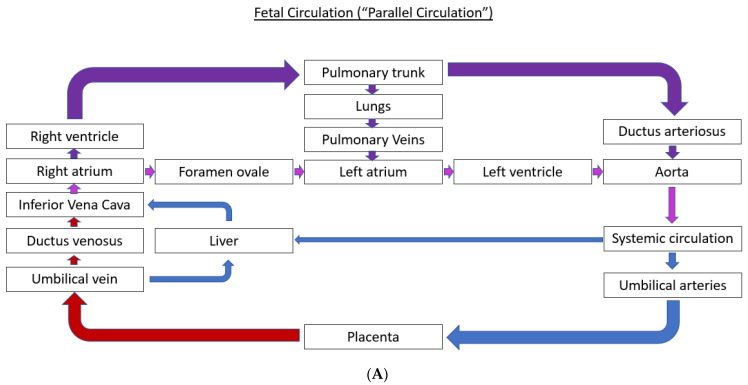
Normal fetal circulation (**A**) and postnatal circulation (**B**). Removal of placenta at the time of birth and subsequent closure of the fetal shunts (ductus arteriosus, ductus venosus and foramen ovale) within a few days causes the transition of fetal circulation to postnatal circulation [14].

**Figure 2 children-10-00281-f002:**
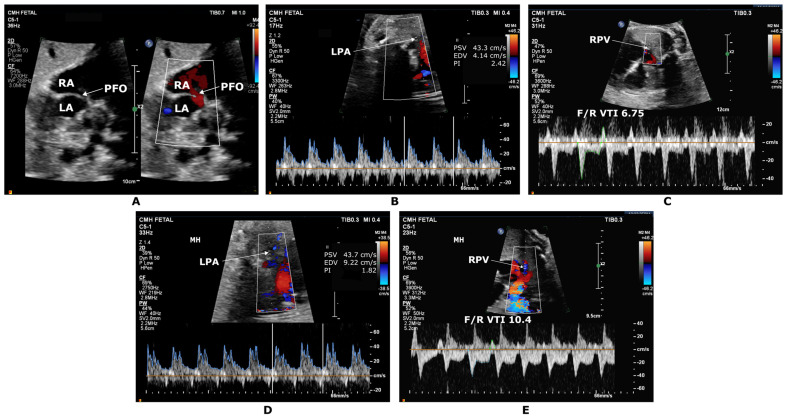
Maternal hyperoxygenation (MH) testing in a third-trimester fetus with hypoplastic left heart syndrome with mitral and aortic stenosis. (**A**) Oblique sagittal color compare image showing a small tunnel-like patent foramen ovale (PFO) with left to right flow. (**B**,**C**) represents the baseline test. (**B**) Left pulmonary artery Doppler (LPA) showing a spiked pattern with pulsatility index (PI) 2.42. (**C**) Right sided pulmonary vein (RPV) Doppler with forward to reverse velocity time integral (F/R VTI) ratio of 6.75. The bottom panel (**D**,**E**) represents post MH testing. (**D**) LPA Doppler broadening with PI 1.82, 24.7% reduction from baseline. (**E**) RPV Doppler showing increase in F/R VTI ratio to 10.4. LA, left atrium; PDA, patent ductus arteriosus; RA, right atrium.

**Figure 3 children-10-00281-f003:**
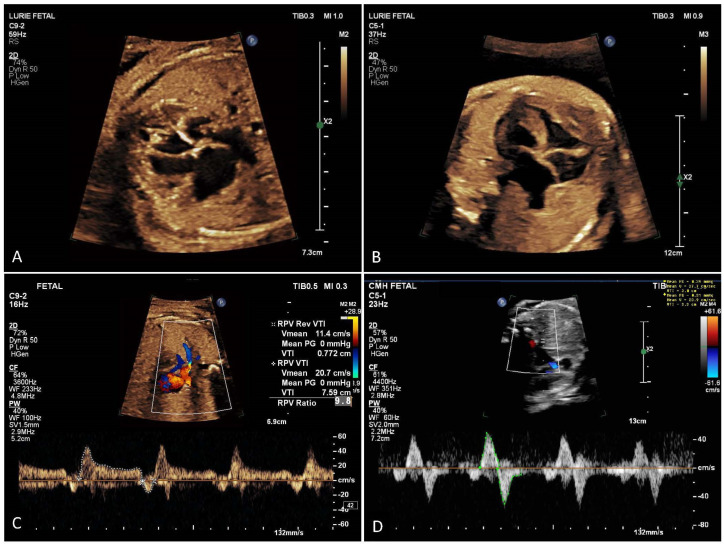
(**A**) Hypoplastic left heart syndrome with widely open atrial septal communication. (**B**) Hypoplastic left heart syndrome with restrictive atrial septum. (**C**) Brief pulmonary venous flow reversal with atrial systole with normal (>5) ratio of forward to reversed flow VTI. (**D**) Significant pulmonary venous flow reversal with abnormal VTI ratio indicating risk of restrictive atrial septum.

**Table 1 children-10-00281-t001:** List of prior studies evaluating the utility of acute maternal hyperoxygenation testing in predicting postnatal hemodynamics in complex congenital heart defects.

Published Studies Author/Year	Fetal Characteristics of the Study Cohort	Fetal Cardiac Diagnoses	MH Protocol	Findings
Rasanen et al. [22] 1998	20 early GA (20–26 weeks) 20 late GA (31–36 weeks)	Healthy fetuses	60% humidified FiO_2_ for 5 min of MH 5 min of recovery	↓ PI in BPA ↑ PI in DA ↓ Foramen ovale flow -Changes are seen only in late GA and not in early GA fetuses -All changes returned to baseline after 10 min of recovery
Szwast et al. [9] 2010	30.1 ± 4.5 weeks GA controls 29.6 ± 5.0	43 HLHS 27 controls	100% FiO_2_ for 10 min via nonrebreather mask at 8 L/min effectively providing 60% inhaled FiO_2_ 5 min of recovery	Reduced pulmonary vasoreactivity (<10% reduction in PI in BPA) correlated with the need for BAS after birth. -No untoward effects seen with MH
Zarkowska-Szaniawska et al. [24] 2011	late gestation	40 fetuses with cardiomegaly and lung hypoplasia	60% FiO_2_ for 15 min	Pulmonary vasoreactivity with MH (>10% reduction in PI in the PA branch) was associated with survival after birth.
Channing et al. [25] 2015	35 ± 3 weeks GA	12 fetuses with an atrial septal aneurysm affecting LV filling and aortic arch flow	100% FiO_2_ for 10 min via nonrebreather mask at 8L/min effectively providing 60% inhaled FiO_2_ 5 min of recovery	MH altered the atrial septal position (↓ atrial septal excursion), improved LV filling, and normalized aortic flow by increasing pulmonary venous return. -Helpful in differentiating small LV due to atrial septal aneurysm vs. true LV hypoplasia or coarctation of the aorta
Enzenberger et al. [11] 2016	>26 weeks GA	22 HLHS	100% FiO_2_ for 10 min	↑ PI in pulmonary veinous Doppler associated with unobstructed atrial septum
Schidlow et al. [12] 2018	>32 weeks GA	2 Ebstein 2 TAPVR 4 HLHS 4 d-TGA	100% FiO_2_ for 10 min at 10L/min effectively providing 60% inhaled FiO_2_ 15 min recovery	Reduced pulmonary vasoreactivity (<20% reduction in PI in PA branches) + cardiac anatomic variables based on the lesion assessed
Rychik et al. [26] 2018 (Abstract only)	35.5 ± 2.4 weeks GA	114 HLHS fetus	100% FiO_2_ for 10 min via nonrebreather mask at 8L/min effectively providing 60% inhaled FiO_2_ 5 min of recovery	No change in Umbilical artery PI (placental resistance unchanged) ↑ cerebral resistance ↓ pulmonary resistance ↑ Ductus arteriosus PI (↑ retrograde flow) No ductal constriction No change in ventricular performance
Mardy et al. [13] 2021	~34 weeks GA	27 HLHS fetuses	100% FiO_2_ for 10 min via nonrebreather mask at 8 L/min effectively providing 60% inhaled FiO_2_ at 8L/min	Poor sensitivity with BPA PI Pulmonary Vein F/R VTI < 6.5, 100% Sensitivity and PPV in predicting emergent atrial septoplasty
Cox et al. [27] 2022	31.0 ± 4.0 weeks for HLHS 27.8 ± 5.1 weeks for controls	9 HLHS 9 controls	100% FiO_2_ for 10 min via nonrebreather mask at 8 L 10 min recovery	↓ LV strain and strain rate (due to ↑ in cerebral vascular resistance) ↑RV strain and strain rate (due to ↓ in pulmonary vascular resistance) ↓ Pulmonary artery PI Most findings did not return to baseline after recovery.

Abbreviations: ↓, Decrease; ↑, Increase; BAS, balloon atrial septostomy; BPA, branch pulmonary artery; DA, ductus arteriosus; d-TGA, d-transposition of the great arteries; GA, gestational age; HLHS, hypoplastic left heart syndrome; LV, left ventricle; MH, maternal hyperoxygenation; PA, pulmonary artery; PI, pulsatility index; FiO_2_, inhaled oxygen; F/R VTI, forward/reverse velocity time integral; RV, right ventricle; TAPVR, total anomalous venous return.

**Table 2 children-10-00281-t002:** Fetal echocardiographic findings and changes with maternal hyperoxygenation as potential predictors of postnatal hemodynamic instability and need for urgent cardiac intervention with various CHDs.

Diagnosis	Baseline Fetal Echocardiogram Findings Suggestive of Hemodynamic Instability after Birth	Expected Changes with MH Performed in the Third Trimester Suggestive of Hemodynamic Instability after Birth	Delivery Room Recommendations
HLHS and variants with severely restrictive or intact atrial septum	Pulmonary vein Doppler [6] − Moderate obstruction: pulmonary vein F/R VTI ratio <5 and >3 [28,29] − Severe obstruction: pulmonary vein F/R VTI ratio < 3 [30]	Reduced pulmonary vasoreactivity − ≤10% reduction in PI* in the branch PAs − Pulmonary vein F/R VTI ratio ≤ 6.5 with MH has 100% sensitivity and specificity [13] for predicting RAS in HLHS	− Initiation of PGE1 infusion − Intubation with mechanical ventilation − Plan for immediate catheter-based or surgical intervention to decompress the left atrium.
TAPVR with significant Obstruction	Pulmonary vein Doppler [6] − Monophasic non-pulsatile pulmonary venous flow − Fetal vertical vein Doppler peak velocity > 0.74 m/s [31]	Mean gradient in the vertical vein after MH correlates with the severity of TAPVR obstruction seen postnatally [12]	− Intubation with mechanical ventilation. − Peripheral IV and/or umbilical line − Initiation of PGE1 infusion (may relax the ductus venosus smooth muscle for infra diaphragmatic TAPVR) − Plan for immediate surgical intervention.
D-TGA and variants with a restrictive atrial septum and prenatal ductal constriction	Abnormal foramen ovale [6,32,33]: − hypermobile septum − the angle of septum primum < 30° − lack of swinging motion of septum or “tethered” septum − bowing of atrial septum > 50% − intact atrial septum Abnormal ductus arteriosus: − small size with moderate/severe restriction reversed, bidirectional or accelerated flow − Abnormal pulmonary vein Doppler − “s” wave velocity > 41 cm/s [33]	Reduced pulmonary vasoreactivity − ≤20% reduction in PI* in the branch PAs − Persistence of bidirectional flow across the foramen ovale	− Initiation of PGE1 infusion through peripheral IV or umbilical line − Intubation with mechanical ventilation − Plan for immediate BAS − If the ductal flow is abnormal and hypoxemia in DR, consider pulmonary hypertension therapy, including intubation, 100% oxygen, iNO
Severe Ebstein anomaly of the tricuspid valve	− Absence of forward flow across the pulmonary valve [34] − Reduced tricuspid regurgitation jet velocity indicating poor RV contractility/systolic function [34] − Flow reversal in ductus arteriosus − Circular shunt physiology prenatally	Pulmonary vasoreactivity with MH > 20% reduction in PI* in the branch PAs and increased cardiac output across the pulmonary valve can predict antegrade flow from the RV to the PA postnatally. The absence of these reassuring findings would be concerning for postnatal hemodynamic instability.	− Peripheral IV or umbilical access − Intubation with mechanical ventilation if needed − Consider 100% oxygen and iNO to decrease pulmonary resistance if there is pulmonary insufficiency (circular shunt) − Consider ECMO − Cardioversion or medical therapy in DR as indicated for arrhythmia
Cardiomegaly and lung hypoplasia	Increased cardiothoracic ratio and concerns for significant lung hypoplasia	Poor pulmonary vasoreactivity with MH (<10% reduction in PI in the branch PAs) associated with non-survivors after birth	− Peripheral IV or umbilical access − Intubation with mechanical ventilation if needed − Consider ECMO

Abbreviations: BAS, balloon atrial septostomy; CHDs, congenital heart defects; DR, delivery room; d-TGA, d-transposition of the great arteries; ECMO, extracorporeal membrane oxygenation; F/R VTI, forward to reverse flow velocity time integral; HLHS, hypoplastic left heart syndrome; iNO, inhaled nitric oxide; IV, intravenous; MH, maternal hyperoxygenation; PAs, pulmonary arteries; PGE1, prostaglandin E1; RAS, restrictive atrial septum; RV, right ventricle; TAPVR, total anomalous pulmonary venous return; TGA, transposition of the great arteries; PI*, pulsatility index = (Peak systolic velocity − end-diastolic velocity)/mean velocity.

## Data Availability

Data availability is not applicable for this review article.

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
