# Peer review of "Utility of Fetal Echocardiography with Acute Maternal Hyperoxygenation Testing in Assessment of Complex Congenital Heart Defects"

_children, 2023, doi:10.3390/children10020281_

Round 1
Reviewer 1 Report
This is overall a well written review of the potential utility of maternal hyperoxygenation in the 3rd trimester for women carrying fetuses with certain forms of congenital heart disease at risk for early neonatal decompensation.
There are some sentences with grammatical errors (example - in the section on Ebstein's anomaly: "The percentage of fetuses who had hemodynamic compromise increases as the pregnancy progresses, indicating a need for serial assessment 41". Please review grammar and punctuation.
In describing the more common protocol the authors stated that most studies administer 100% FiO2 while Table 1 shows that 5 studies used 60% and 4 studies used 100%- so it may be more accurate to state that more recent studies have administered 100% oxygen.
Table 2. Summarizes what the authors expect to see in MH for each type of lesion described. This is based on the literature listed in the prior table 1, however, there are minimal citations in this table and these are findings from a few studies without many subjects, so citation is important. In addition, due to still relatively little data on this subject, I would suggest the description of this Table (2) be something like "Potential fetal Echocardiographic findings and changes with maternal hyperoxygenation..." as there still is a lack of evidence for several of these lesions and these ideas are currently still being investigated. Also, for Table 2, the thrid column should state that the MH is given in the 3rd trimester.
Figure 1- Fetal echo images and Dopplers are quite small and should be made larger for the reader to understand the precise location of Doppler acquisition.
Finally, with regard to safety in MH, though the authors report no evidence of ill effects on mother or fetus thus far, they should acknowledge that there is still a lack of data on this and that some studies have shown an increase in cerebral resistance with MH- it is listed in Table 1 (Rychik abstract) but never discussed in the safety section. Other studies have reported changes in cerebral perfusion with MH and this deserves some mention.
Author Response
Dear Reviewer,
Thank you for your thoughtful review and suggestions. We have made changes to the manuscript according to your feedback as described below in details:
Line by line responses:
This is overall a well written review of the potential utility of maternal hyperoxygenation in the 3rd trimester for women carrying fetuses with certain forms of congenital heart disease at risk for early neonatal decompensation.
Comment # 1
There are some sentences with grammatical errors (example - in the section on Ebstein's anomaly: "The percentage of fetuses who had hemodynamic compromise increases as the pregnancy progresses, indicating a need for serial assessment 41". Please review grammar and punctuation.
Author Response:
We have made grammatical edits to the manuscript. The manuscript was reviewed and edited for grammatical accuracy by the medical writing team at the UKMC School of Medicine and we have acknowledged their contribution at the end of the manuscript.
Comment # 2
In describing the more common protocol the authors stated that most studies administer 100% FiO2 while Table 1 shows that 5 studies used 60% and 4 studies used 100%- so it may be more accurate to state that more recent studies have administered 100% oxygen.
Author Response:
We agree with this suggestion. We have made clarifications in the manuscript indicating that most of the studies (7 out of 9) used 100% oxygen given via a non-rebreather face mask which ultimately provides 60% inhaled oxygen. We have edited this information, further clarifying the administered oxygen concentration (100%) to the mother via non-rebreather face mask, which provides 60% inhaled oxygen to the mother due to the non-rebreather mask technology. We have clarified this information throughout the manuscript.
Changes made to the manuscript:
“Based on the MH protocols used in prior studies, 100% humidified oxygen is typically delivered to the mother using a non-rebreather mask which provides around 60% inhaled oxygen concentration to the mother for 10 min before reevaluating select fetal echocardiographic measures."
Comment # 3
Table 2. Summarizes what the authors expect to see in MH for each type of lesion described. This is based on the literature listed in the prior table 1, however, there are minimal citations in this table and these are findings from a few studies without many subjects, so citation is important. In addition, due to still relatively little data on this subject, I would suggest the description of this Table (2) be something like "Potential fetal Echocardiographic findings and changes with maternal hyperoxygenation..." as there still is a lack of evidence for several of these lesions and these ideas are currently still being investigated. Also, for Table 2, the thrid column should state that the MH is given in the 3rd trimester.
Author response:
We agree with these suggestions.
- We have added citations to Table 2.
- The title of Table 2 has been edited as following.
“Table 2 Fetal echocardiographic findings and changes with maternal hyperoxygenation as potential predictors of postnatal hemodynamic instability and need for urgent cardiac intervention with various CHDs."
- We have added the information that the MH is given in the third trimester.
Table 2 Column 3 heading: “Expected changes with MH performed in the third trimester suggestive of hemodynamic instability after birth.”
Comment # 4
Figure 1- Fetal echo images and Dopplers are quite small and should be made larger for the reader to understand the precise location of Doppler acquisition.
Author Response: We have provided high-resolution images and the location of the Doppler acquisition is indicated by an arrow.
Comment # 5
Finally, with regard to safety in MH, though the authors report no evidence of ill effects on mother or fetus thus far, they should acknowledge that there is still a lack of data on this and that some studies have shown an increase in cerebral resistance with MH- it is listed in Table 1 (Rychik abstract) but never discussed in the safety section. Other studies have reported changes in cerebral perfusion with MH and this deserves some mention.
Author response:
We agree with these suggestions and have expanded the description related to the safety of MH testing.
Changes made to the manuscript:
This study reported a temporary increase in cerebral resistance; however, the changes were temporary. These findings suggested that when MH is used as a provocative test, by giving oxygen for a short period as a diagnostic aid, there were no significant adverse events based on the maternal clinical parameters or fetal echocardiography Doppler parameters, indicating that this test is safe for the mothers and fetuses 10. Therefore, in many centers, MH is no longer considered an experimental test and is used as part of a clinical protocol. Although data from these prior studies remain limited and further studies in this area would be beneficial in ensuring the safety and utility of MH testing.
Reviewer 2 Report
Here are some questions and recommendations,
1. It will be better to use a figure to illustrate circulation changes before and after birth, and how MH can have the effect to the cardiovasculature of the fetus.
2. Figure 2. It’s better to indicate the major arteries, the pulmonary venous flow direction with arrows in the pictures.
3. The authors mentioned that acute MH testing had been investigated for 5 decades, but if the importance is significant, why is there still no standard protocol for this method? Please indicate in the text.
4. Besides MH plus echo, are there other methods that can be used to predict the risk of hemodynamic instability for patients with congenital heart disease after immediate birth? What are the advantages and disadvantages by comparisons with these methods?
Author Response
Dear Reviewer,
Thank you for your thoughtful review and suggestions. We have made changes to the manuscript according to your feedback as described below in details:
Line by line responses are provided below and please also see the attachment which includes the same responses with figures.
Comment # 1.
It will be better to use a figure to illustrate circulation changes before and after birth, and how MH can have the effect to the cardiovasculature of the fetus.
Author response:
We appreciate this suggestion. We have included a previously published diagram (Figure 1) describing the fetal and postnatal circulation with appropriate citations referring to the original publication.
Figure 1: Normal fetal circulation (Image 1A) and postnatal circulation (Image 1B)
Figure 1 DA, Ductus arteriosus; DV, Ductus venosus, FO, Foramen ovale; IVC, Inferior vena cava; LA, left atrium; LV, left ventricle; PA, pulmonary artery; PV, pulmonary vein; R, resistance; RA, right atrium; RV, right ventricle; SVC, superior vena cava; UA, umbilical artery; UV, umbilical vein. Image taken from Sadler et al “Langman’s Medical Emryology” 2006
Comment # 2.
Figure 2. It’s better to indicate the major arteries, the pulmonary venous flow direction with arrows in the pictures.
Author response: We agree with the suggestion and have edited the figures to include arrows indicating the flow direction.
Comment # 3.
The authors mentioned that acute MH testing had been investigated for 5 decades, but if the importance is significant, why is there still no standard protocol for this method? Please indicate in the text.
Author response:
Although acute MH testing has been investigated for five decades, its utility, along with fetal echocardiography, is only evaluated more recently. Therefore, there is no standard protocol yet. We hope that this review article will help future centers develop MH programs in their centers and that more data will help create a standard protocol.
Changes made in the manuscript:
“However, the cardiac and circulatory changes with MH were not easy to evaluate prior to the availability of fetal echocardiography in the late 1980s."
Comment # 4.
Besides MH plus echo, are there other methods that can be used to predict the risk of hemodynamic instability for patients with congenital heart disease after immediate birth? What are the advantages and disadvantages by comparisons with these methods?
Author response:
Table 2 summarizes fetal echocardiographic findings that can predict the risk of hemodynamic instability (column 2). However, as described in the background, fetal echocardiographic measures may underestimate the risk of postnatal hemodynamic instability due to the differences in fetal and postnatal circulation. Besides baseline fetal echocardiography listed in column 2 and the potential findings with MH testing listed in column 3, there are no other methods that are currently used to predict the risk of hemodynamic instability with CHD after birth.

Reviewer 3 Report
Congratulations for the review on maternal hyperoxygenetation. Based on your review I would recommend one additional table: when MH should be offered (at what gestational age) and when it is too early.
Your review of d-TGA and postnatal baloon atrispetostomy is incomplete - couple of papers are missed, for instance: Słodki M, Axt-Fliedner R, Zych-Krekora K, Wolter A, Kawecki A, Enzensberger C, Gulczyńska E, Respondek-Liberska M; International Prenatal Cardiology Collaboration Group. New method to predict need for Rashkind procedure in fetuses with dextro-transposition of the great arteries. Ultrasound Obstet Gynecol. 2018 Apr;51(4):531-536. doi: 10.1002/uog.17469. PMID: 28295809..
Fig.2D is not the best one, I would suggest to replace it
Author Response
Thank you for your thoughtful review and suggestions. We have made changes to the manuscript according to your feedback as described below in details:
Line by line responses:
Congratulations for the review on maternal hyperoxygenetation. Based on your review I would recommend one additional table: when MH should be offered (at what gestational age) and when it is too early.
Comment # 1
Your review of d-TGA and postnatal baloon atrispetostomy is incomplete - couple of papers are missed, for instance: Słodki M, Axt-Fliedner R, Zych-Krekora K, Wolter A, Kawecki A, Enzensberger C, Gulczyńska E, Respondek-Liberska M; International Prenatal Cardiology Collaboration Group. New method to predict need for Rashkind procedure in fetuses with dextro-transposition of the great arteries. Ultrasound Obstet Gynecol. 2018 Apr;51(4):531-536. doi: 10.1002/uog.17469. PMID: 28295809..
Author response:
We appreciate this feedback and the suggested reference is included as a text and in Table 2.
Changes made in the manuscript:
Additionally, pulmonary venous maximum velocity of > 41 cm/s was a helpful predictor of postnatal need for BAS in a study by Slodki et al.
Comment # 2
Fig.2D is not the best one, I would suggest to replace it
Author response:
We appreciate the feedback, and the figure has been replaced with high resolution image and markings to identify cardiac structures within the image.